# Eyes Are the Windows to the Soul: Reviewing the Possible Use of the Retina to Indicate Traumatic Brain Injury

**DOI:** 10.3390/ijms26115171

**Published:** 2025-05-28

**Authors:** Loretta Péntek, Gergely Szarka, Liliana Ross, Boglárka Balogh, Ildikó Telkes, Béla Völgyi, Tamás Kovács-Öller

**Affiliations:** 1Szentágothai Research Centre, University of Pécs, H-7624 Pécs, Hungary; pentekloretta00@gmail.com (L.P.); gergely.sz@gmail.com (G.S.); boglarka.balogh311@gmail.com (B.B.); volgyi01@gamma.ttk.pte.hu (B.V.); 2Department of Neurobiology, University of Pécs, H-7624 Pécs, Hungary; 3NEURON-066 Rethealthsi Research Group, H-7624 Pécs, Hungary; 4Department of Neuroscience, University of Calgary, Calgary, AB T2N 1N4, Canada; liliana.ross@ucalgary.ca; 5Medical School, University of Pécs, H-7624 Pécs, Hungary; ildiko.telkes@aok.pte.hu

**Keywords:** traumatic brain injury, retina, cellular marker, microglia, visual system, imaging, disease

## Abstract

Traumatic brain injury (TBI) induces complex molecular and cellular responses, often leading to vision deterioration and potential mortality. Current objective diagnostic methods are limited, necessitating the development of novel tools to assess disease severity. This review focuses on the retina, a readily approachable part of the central nervous system (CNS), as a potential indicator of TBI. We conduct a targeted database search and employ a blinded scoring system, incorporating both human and artificial intelligence (AI) assessments, to identify relevant articles. We then perform a detailed analysis to elucidate the molecular pathways and cellular changes in the retina following TBI. Recent findings highlight the involvement of key molecular markers, such as ionized calcium-binding adapter molecule 1 (IBA1), phosphorylated tau, glial fibrillary acidic protein (GFAP), and various cytokines (IL-1β, IL-6, and TNF). Additionally, the roles of oxidative stress, reactive oxygen species (ROS), and blood–retina barrier (BRB) disruption are explored. Based on these findings, we hypothesize that alterations in these molecular pathways and cellular components, particularly microglia, can serve as direct indicators of brain health and TBI severity. Recent technological advancements in retinal imaging now allow for a direct assessment of retinal cells, including microglia, and related inflammatory processes, facilitating the translation of these molecular findings into clinical practice. This review underscores the retina’s potential as a non-invasive window into the molecular pathophysiology of TBI.

## 1. Introduction

The phrase “eyes are the windows to the soul” means that one can infer significant insights into a person’s thoughts and feelings by looking into their eyes. While this phrase is often used in literature, in our context, we use it metaphorically to explore the potential of utilizing the retina to reveal TBI and to assess the severity of brain injuries.

There is no perfect marker to identify TBI severity to date. Additionally, there is a need for non-invasive techniques that improve the assessment of disease severity.

TBI is increasingly known as a significant contributor to retinal dysfunction, which is not surprising considering the shared physiological and developmental characteristics between the retina and the CNS. The impact that TBI has on retinal integrity manifests through inflammatory responses, neurodegenerative processes, oxidative stress, and functional impairments, which collectively suggest the retina as a potential biomarker for assessing TBI severity and progression. Here, we aim to sum up the recent literature about brain–retina relations in TBI to identify novel details for identifying TBI severity.

## 2. Methods

### 2.1. Searching, Scoring, and Processing of Articles

To identify articles relevant to this review, we first used the following search terms in the NCBI database (NIH, USA): (retina AND (traumatic brain injury [Title/Abstract]). Based on this, we obtained 142 articles in the first round of search. Following this, we used a 1–10 scoring system (1–3: non-relevant, 4–6: fair relevance, 7–9: very relevant, and 10: perfect match to the topic) on the titles and abstracts. Four authors independently scored all articles based on the titles and abstracts. We used Copilot (Microsoft) as a fifth scorer. The AI model was limited to testing only 13–15 articles after testing scoring reliability based on divergence (error rate > 4 human mean error) from the human scores; therefore, we introduced all articles in 12 packages. This resulted in an acceptable score with low divergence from the human mean. The average scores (4 humans and 1 AI) of 66 searched papers were above the median score. We included these in our detailed reading and analysis. These articles were equally shared between 5 human authors (AI was only used for scoring) after blind selection, and the relevant information was added to the pre-defined paragraphs in an ad hoc manner. Subsequently, correct phrasing, information linking, and writing were carried out by all authors on a shared Google Drive platform. Additionally, 6 other articles were included in the Discussion to aid the understanding of recent developments and how these can come together to aid the translation of our findings from the literature search (see the detailed process in a flowchart in Figure 1).

### 2.2. Figures

Figures were constructed with ibisPaint X (Version 13.0.4) and MS PowerPoint based on the literature.

## 3. Manifestations of TBI in Patients

Brain trauma (TBI) is often caused by an impact to the skull, but due to the stochastic divergence of non-dissipated pressure changes, the effects can vary significantly, ranging from changes in the level of molecular markers to the manifestation of distinct symptoms (Figure 2).

Mouse models have demonstrated various behavioral manifestations of TBI, including increased light aversion, reduced contrast sensitivity in the optokinetic reflex test, and enhanced pupil contraction in the pupillary light response test [2], symptoms similar to those observed following ocular blast injury [6]. Photophobia, an increased sensitivity to light, was reported after TBI and was hypothesized to result from an increased sensitivity of intrinsically photosensitive retinal ganglion cells (ipRGCs) due to injury [3,4]. However, the same study reported no significant differences in pupillary constriction or re-dilation amplitudes in response to blue or red light between photophobic participants with mild TBI and controls, suggesting no increased ipRGC light sensitivity [4]. Interestingly, however, TBI participants exhibited more variable pupil responses to blue light than controls, indicating that an alteration in ipRGCs distinct from light sensitivity may underlie TBI-induced photophobia [4].

As revealed before, a significant number of blast-induced TBI patients suffer from visual impairments, with prevalence rates varying based on injury severity and the specific visual function [7]. Among them, hemianopia following TBI is relatively common and occurs due to retrochiasmatic damage. Homonymous hemianopia may result from a lesion affecting the contralateral optic tract (OT), lateral geniculate nucleus (LGN), optic radiation, or primary visual cortex [5].

Beyond the immediate effects, multiple studies indicate a link between previous TBI and a higher occurrence of Alzheimer’s disease (AD), Parkinson’s disease, amyotrophic lateral sclerosis, cognitive impairment, and multiple sclerosis, pointing to ongoing neurodegeneration following primary and secondary TBI [8]. In the United States, only 3% (~51,000) of all TBI cases (1.7 million annually) result in death [9], but nearly one-third of injury-related deaths originate from TBI, with an even higher prevalence (up to 85%) of vision impairment among all cases. Among military personnel, blast-mediated TBI (bTBI) injury is the most prevalent, although it is considered a distinct type of injury caused by a pressure wave [10].

TBI symptoms can include double vision or diplopia, asthenopia with blurred vision, reading difficulties, restricted peripheral vision, a loss of place, photosensitivity, and changes in color perception. The severity of TBI can affect the outcome, with severe TBI (sTBI) bearing the most injuries with multiple lesions and more pronounced inflammatory effects, while moderate and mild TBI can still cause visual dysfunctions. Repetitive mild injuries can lead to the loss of RGCs in animal models, but human-related studies are lacking. Some studies differentiate traumatic optic neuropathy (TON) related to TBI if direct deterioration and accumulated visual symptoms are observed, although the diagnosis is quite challenging [11]. TBI includes primary and secondary injuries in the CNS. Primary injury is caused by a physical force impacting the skull and the brain, while secondary injury can occur later, beginning a few minutes following the impact and lasting for several weeks. It comprises edema, blood–brain barrier (BBB) breakdown, ROS release, calcium ion imbalances, and inflammation. Silver staining indicates the maximal peak in tissue degeneration 48 h after injury, continuing for up to 7 days; however, these numbers are just rough approximations due to the post-mortem applicability of the approach [12]. The damage induced by neutrophils also spikes 12–48 h after experimental TBI [13].

Even in mild TBI, patients display long-term cognitive impairment, neuropsychiatric symptoms, and post-traumatic stress disorder [9]. In patients, the occurrence of post-chiasmal field defects ranged from 3.2% to 39% in TBI [9]. CT and MRI scans are the most frequently employed brain imaging technologies for TBI diagnosis but have obvious limitations, including low sensitivity and resolution, invasive nature, or limited access to equipment [14].

The strong connection between visual deficits and TBI can be easily assessed using a symptom questionnaire, the Brain Injury Vision Symptom Survey (BIVSS), to document vision deficits and distinguish between TBI severity levels [14]. Evidence from multiple studies highlights the prevalence of visual symptoms and light sensitivity in TBI patients. A retrospective analysis of 100 TBI cases indicated that approximately 50% of these patients presented with visual disturbances. Complementary case–control studies have corroborated this, showing that photophobia is four times more common in TBI patients than in healthy controls and that TBI patients report significantly more intense light-induced headaches [15].

Both single and repeated injury protocols led to the death of retinal ganglion cells (RGCs) and the deterioration of the optic nerve, with these effects being linked to the impact magnitude and the number and frequency of injuries. Phosphorylated tau immunoreactivity was not present in the brains of animals exposed to repetitive mild TBI (rmTBI). Therefore, the above-listed evidence indicates that retinas in rmTBI-induced traumatic axonal injuries (TAIs) may serve as new diagnostic tools, particularly those affecting the visual system and/or cerebellum [16].

## 4. Effects of TBI on the Visual System

The retina, as part of the CNS, is particularly susceptible to damage, including that induced by TBI (Figure 3). The literature describes different TBI types and models. For clarification, we describe the models or types of TBI if needed, as they can have minor but important differences in their effects on the visual system.

Studies show that blast-induced trauma reduces photoreceptor cell number, decreasing the thickness of the outer nuclear layer of the retina [6,17]. Additionally, blast-induced TBI and repeated ocular blast injuries cause optic nerve axon loss, correlating with reduced contrast sensitivity [2,6].

In a central fluid percussion injury model of TBI, researchers observed widespread optic nerve TAI. Interestingly, however, in this model, TAI did not result in the loss of RGCs. Instead, the distal optic nerve segments showed degeneration, marked by increased microglia and macrophage activity, while proximal axonal segments remained intact, preserving RGC integrity [18]. This result may have occurred because the fluid percussion injury model of TBI causes brief optic nerve deformation near the optic chiasm instead of closer to the RGC layer of the retina [18]. ipRGCs were also affected, showing a decreased soma size and increased melanopsin immunolabeling following blast-induced TBI [2]. Conversely, the density of displaced amacrine cells remained unaffected [19], indicating that the observed changes were induced in ganglion cells via their axons.

Functional impacts on the visual system were evident in electroretinogram (ERG) measurements, which revealed decreased A- and B-wave peak amplitudes following blast-induced TBI [2]. In mice, pattern ERG (PERG) assessments of RGC function showed an initial decrease in response amplitude one-week post-injury, a temporary recovery at four weeks, and a subsequent decline by sixteen weeks [19,20]. However, under conditions of elevated ocular pressure, achieved by tilting the TBI mice to a 60-degree head-down position during PERG testing, a decreased voltage change amplitude was detected at four weeks post-injury. This aligns with a reduced RGC layer thickness and increased RGC spontaneous activity throughout the testing period, suggesting alterations in results due to measurement changes [19].

Human TBI studies showed some important changes in the retina. Henle’s fiber layer (HFL) is made up of Müller cells and photoreceptor axons extending diagonally from their nuclei to synapse with inner nuclear cells. Because of their directional reflectivity, they can be distinguished from the optical outer nuclear and outer plexiform layers close to the fovea. In individuals with a history of TBI, HFL was found to be thicker than in controls, potentially due to changes in the deep capillary plexus structure or an increased Müller cell volume [21]. However, global HFL phase retardation remained unchanged [21]. Additionally, the ERG A-wave amplitude was reduced, indicating impaired photoreceptor function [21].

Early diffusion tensor imaging (DTI) can identify OT lesions causing homonymous hemianopia. DTI also detects diffuse changes in functionally intact structures that resolve over time, likely due to minor diffuse axonal injury recovery. Optical coherence tomography (OCT) reveals retinal thinning consistent with Wallerian degeneration of the optic tract. However, since OT atrophy can also stem from LGN or primary visual cortex lesions, OCT lacks precise localization value [5].

In another study, OCT revealed a temporal peripapillary retinal nerve fiber layer (pRNFL) thinning linked to TON in patients with chronic mTBI [22].

A chronic decline in the optomotor response was observed five months post-blast TBI [20]. Mild TBI leads to widespread axonal damage and microglial activation, with injured axons releasing damage-associated molecular patterns (DAMPs), including ATP and S100b. These molecules bind to microglial receptors, triggering a pro-inflammatory response that exacerbates damage [23].

TON and retinal thinning are distinct forms of injury that appear to contribute to TBI effects in the retina. TON from blunt force trauma affects 0.5–5% of TBI cases, resulting in irreversible vision loss [12]. In studies on mice and Olympic boxers, RNFL thinning correlated with optic nerve thinning [9]. An ultrasonographic assessment of the optic nerve sheath diameter (ONSD) can help in the identification of increased intracranial pressure (ICP) in TBI patients, showing a near-linear correlation with invasive methods [24].

Repetitive mild TBI led to more profound RNFL thinning than single-blast injury, likely due to the short interval (48 h) between impacts, which prevented optic nerve edema from subsiding [25]. In mice subjected to blast-induced TBI, vitreous detachment, hemorrhage, subretinal bleeding, and photoreceptor degeneration were observed [26]. Complement C3 was deposited in the retinogeniculate synapses of the dorsal LGN (dLGN) 3 days post-TBI, with microglial changes persisting for up to 49 days. Complement inhibitor CR2-Crry countered these effects, promoting neuron survival [27]. rmTBI led to visual pathway dysfunction beyond the retina, indicated by severely diminished visual-evoked potentials (VEPs) and unchanged ERGs. This suggests that TON and widespread axonal injury are contributors to visual dysfunction [28]. Extensive microglial activation was noted at TAI sites, particularly in the OT, corpus callosum, and superior colliculus [16]. Patients with migraine-related photosensitivity exhibited light-triggered migraine attacks via the ipRGC pathway [29]. In mTBI, cone response reduction and rod pathway overactivation at high light levels may explain photalgia. This phenomenon likely results from the loss of cone-generated inhibition, leading to ERG shifts toward rod-like properties [30].

## 5. Cellular and Molecular Markers of TBI in the Retina

TBI-induced pathologies extend beyond the location of the primary insult in the CNS and manifest as distinct morphological and functional changes in the retina. These changes are accompanied by alterations in key retinal cellular markers, a comprehensive list of which is provided in the paragraphs below (Table 1).

Glial activation and intermediate filament (IF) remodeling occur in the retina as part of TBI. The increased presence of GFAP immunoreactivity in the mouse retina post-TBI or acoustic blast overpressure (ABO) activates Müller glia [17,31,32]. However, GFAP is not the sole IF affected; other type III IFs like vimentin and desmin, as well as type VI IFs such as synemin and nestin, are also remodeled or induced in the rat retina following ABO exposure [32]. The resulting glial phenotype depends on the specific retinal region [32]. Similarly, increased IBA1 immunoreactivity reflects the activation of microglia and a TBI-induced inflammatory response [2,17,31]. Pro-inflammatory microglial activation is further evidenced by elevated CD68 expression, indicating phagocytic activity [17].

There is a strong link between Tau phosphorylation and neurodegeneration. Phosphorylated tau, a hallmark of neurofibrillary tangles in neurodegenerative diseases, is elevated in retinal horizontal cells and Müller glia post-TBI [17]. Here, the release of lysophosphatidic acid (LPA) from activated microglia, astrocytes, and platelets can induce further inflammatory processes and tau phosphorylation [33].

Cytokines and oxidative stress work hand in hand to alter cellular survival. An increase in the retinal levels of cytokines, including interleukin (IL)-1a, IL-1b, IL-6, and tumor necrosis factor, occurs post-blast-induced TBI [31]. Blocking the IL-1 pathway with anakinra (an IL-1 receptor antagonist) reduces glial activation, preserves RGC signaling, and safeguards the structural integrity of RGCs and the optic nerve [31].

TBI is also associated with an overexpression of ROS, which may contribute to retinal pathologies [34]. ROS overexpression in RGCs triggers the upregulation of Kruppel-like factor 4 (KLF4), activating pro-apoptotic p53 signaling while inhibiting pro-survival STAT3 signaling [34].

TBI leads to BBB and BRB disruption, allowing peripheral immune cell infiltration [12]. Infiltrating cells, including macrophages and neutrophils, exacerbate inflammation and contribute to RNFL thinning [12].

The activation of inflammatory pathways and the disruption of barriers (the BBB and BRB) overwhelms the CNS and induces secondary processes. The CCL20/CCR6 axis, known for modulating inflammation, plays a critical role in neurodegeneration post-TBI, cerebral ischemia, and spinal cord injury. Elevated CCL20 levels are detected in human plasma 24 hours after sTBI [35].

Importantly, apoptosis and microglial activation, together with an accelerated immune response, can be widely detected in the retina and in the brain. In a severe and repetitive mild TBI model, Caspase-3 activation signals apoptotic events and axonal deterioration [36]. Interestingly, while microglia avoid apoptotic induction, astrocytes express activated Caspase-3, an early marker of retinal damage [36].

Complement C3 is deposited in retinogeniculate synapses post-TBI, where CR2-Crry complement inhibition shows a neuroprotective effect [27]. Additionally, dendritic cells from Rag1^-/-^ mice may present RGC antigens, eliciting an immune response in wild-type mice [20].

A robust phagocytic response, detected with CD68 immunohistochemistry, is linked to traumatic axonopathy. However, in Sarm1 knockout mice, the suppression of CD68 activation suggests a potential direct role of SARM1 in microglial activation [37].

Increased levels of β-amyloid and the lipid peroxidation product 4-hydroxy-trans-2-nonenal (4HNE) are detected in the retina acutely post-blast injury, consistent with the elevated oxidative stress markers in human plasma [38].

In repetitive diffuse mTBI models, tauopathy is accelerated in genetically susceptible mice [39]. In α RGCs, downregulating phosphatase and tensin homolog (PTEN) promotes regeneration, further enhanced by osteopontin and insulin-like growth factor 1 [40].

**Table 1 ijms-26-05171-t001:** Cellular and molecular retinal markers of TBI.

Marker	Function	Study
GFAP (glial fibrillary acidic protein)	Indicates Müller glia activation in the retina post-TBI or acoustic blast overpressure	[17,31,32]
IBA1 (ionized calcium-binding adapter molecule 1)	Reflects microglial activation and inflammation post-TBI	[2,17,31]
CD68 (Cluster of Differentiation 68)	Marker of pro-inflammatory microglial activation and indicates phagocytic response in traumatic axonopathy	[17,37]
Phosphorylated tau	Indicates neurodegeneration; associated with neurofibrillary tangles	[17]
LPA (lysophosphatidic acid)	Induces inflammatory processes, astrocyte proliferation, and tau phosphorylation	[33,41,42]
IL-1B, IL-1a, IL-6, TNF (interleukin; tumor necrosis factor)	Cytokines involved in inflammation and oxidative stress post-TBI	[31]
KLF4 (Kruppel-like factor 4)	Inhibits pro-survival STAT3 and triggers pro-apoptotic p53 in RGCs	[34]
CCL20 (chemokine (C-C motif) ligand 20)	Involved in neurodegeneration and inflammation post-TBI	[35]
Caspase-3	Marker of apoptosis, expressed in astrocytes in retinal damage	[36]
Complement C3	Deposits in retinogeniculate synapses post-TBI; inhibition is neuroprotective	[27]
β-amyloid and 4HNE (4-hydroxy-trans-2-nonenal)	Oxidative stress markers post-TBI	[38]
PTEN (phosphatase and tensin homolog)	Downregulation promotes regeneration of α RGCs	[40]
Osteopontin and IGF-1 (insulin-like growth factor 1)	Enhances RGC regeneration	[40]

## 6. Possible Translational Implementation of Retinal Markers in TBI

### 6.1. The Importance of Early TBI Detection

The effects of TBI can be detected in the retina, even in cases where the injury’s impact is insufficient to produce observable alterations in the brain. For example, even in cases where blast-induced trauma failed to increase GFAP, IBA1, and phosphorylated tau immunoreactivity in the prefrontal cortex, levels of these damage and inflammatory markers were elevated in the retina [17]. This suggests that the retina is more susceptible to TBI and can therefore serve as a useful indicator of injury following blast exposure [17]. In many life situations where TBI is common, including the military and sports, patients would greatly benefit from the ability to detect TBI early via non-invasive assessments of retinal structure and/or visual function. Furthermore, the reliable detection of TBI through assessments of the visual system would also have implications for legal frameworks, particularly in cases of suspected abusive head trauma (AHT) in children [43]. For example, retinal hemorrhages involving multiple layers of the retina and extending to the ora serrata have been observed to occur more frequently in cases of AHT than in non-abusive head trauma [43].

### 6.2. Methods for Early TBI Detection

To effectively use visual system symptoms as indicators of TBI, it is imperative to develop reliable and non-invasive tests for the early detection of visual deficits.

It is known that, for individuals with a genetic predisposition to amyloidosis in AD, exposure to blast TBI accelerates damage to RGCs and their axons. This damage is linked to a low-level increase in Aβ pathology of the brain. These findings suggest that genetic risk factors for AD may heighten the retina’s vulnerability to blast-induced injury. A study demonstrated the long-term effects of blast injury on both retinal function and structure, as well as amyloid accumulation. Future translational biomarker research could focus on integrating retinal function assessments with brain amyloidosis imaging using positron emission tomography (PET). This approach is particularly valuable for aging veterans who suffered blast TBI [44].

The Centers for Disease Control and Prevention (CDC), USA, reports that moderate TBI individuals are 2.3 times more likely to develop AD, attributing this risk to the persistent neuroinflammation following TBI. This is particularly significant, as research suggests that TBI may accelerate the onset of AD pathology by 4 to 10 years. Given that AD pathogenesis involves similar disruptions in cellular homeostasis, microglial and astrocyte activation, and inflammatory processes, the impact of trauma on the CNS is a compelling area of study [45]. Nerve sheath diameter is linked to retinal health through its ganglion cell origin. As RGCs are affected by either direct eye trauma or TBI, deterioration of the axons can be measured with ultrasound imaging [24]. Another important study investigating TBI-induced RNFL thinning found 30% of eyes to have significant RNFL thinning and 40% to have visual function defects 6 months following mTBI, at which point visual function deficits and the structural changes were the most extensive. Scatter visual field impairment was among the most commonly reported deficits with a strong, significant association with RNFL thinning [46]. Due to the strong connection between cerebral blood flow and retinal blood flow, non-invasive ocular imaging of retinal blood flow can be used to pinpoint TBI severity. Related to blood flow, retinal oximetry can actively determine the oxyhemoglobin/deoxyhemoglobin ratio and oxygen utilization in the retina [9].

Subtle spectral changes were detected with Raman spectroscopy, likely associated with a decrease in cardiolipin expression and indicating metabolic disruption from the brain and retina following injury [40,47].

PERG may be a useful tool, as it can assess RGC function non-invasively. In fact, it has previously been shown that marked PERG signal changes occur in mice following blast-induced TBI [19]. It appeared that performing PERG measurements under conditions of increased ocular pressure resulted in differences in retinal function that can be detected by such measurements [19]. This suggest that PERG can have high variability and requires extensive normalization.

In veterans with blast-related TBI, automated perimetry measurements have proven to be a practical and effective tool for detecting visual field deficits when certain conditions are met, including testing during a time of maximal alertness and the absence of sensorineural medications. Additionally, 15% of participants showed significant visual field defects, such as hemianopia or quadrantanopia, while a further 36% showed abnormal global visual field indices. These results suggest that there is a high prevalence of visual field deficits among individuals with blast-related TBI [7].

There is another method, called multifocal pupillographic objective perimetry (mfPOP), that can also provide reliable results. Athletes who recently suffered mild TBI were studied, and their visual response amplitude showed a significant decrease compared to that of control subjects. Furthermore, a significant negative correlation was found between the thickness of the retina and the latency of the mfPOP response in athletes with acute mTBI. This indicates that structural changes in the retina after injury may be associated with functional visual impairment. The results here should also be treated with caution, considering that various antidepressants, decongestants, and other systemic diseases can affect pupil size [48].

That said, microglia show great promise in detecting the early signs of TBI in a multitude of models [26], but this has never been linked to the option of using them as an additional marker for categorizing TBI severity.

### 6.3. Possible Interventions for Early TBI Treatment

In terms of possible interventions to reduce TBI-induced retinal pathologies, the intravenous administration of an anti-LPA antibody 1 h after blast exposure improved rat visual acuity, improved retinal signaling function, and preserved neuronal cell integrity following blast exposure [33]. Furthermore, in contrast to antibody-treated rat controls, the anti-LPA-treated counterparts did not show an increase in GFAP-positive activated Müller cells eight days after blast exposure [33]. Ultimately, the administration of anti-LPA antibody immediately post-blast exposure seems effective at reducing the pro-inflammatory effects of LPA described earlier.

Another possible therapeutic target to reduce TBI-induced retinal pathologies is KLF4. Suppressing KLF4 significantly boosts RGC axon regeneration, especially when stimulated by ciliary neurotrophic factor (CNTF) after optic nerve crush [34]. This finding suggests that blocking the action of KLF4 following TBI-induced tissue trauma may serve to enhance tissue repair and thus attenuate retinal pathologies.

Alternatively, since blocking the interleukin pathway with the IL-1 receptor antagonist anakinra was found to reduce glial activation and RGC and optic nerve damage following blast-induced TBI, this suggests that retinal inflammatory pathways may be effective therapeutic targets to attenuate TBI-induced retinal pathologies [31].

There is also evidence that cannabinoid receptor type 2 (CB2) reverse agonist treatment up to 2 days following TBI can also help mitigate visual system deficits [2]. Specifically, intraperitoneal raloxifene injections up to 48 h post-blast-induced TBI appeared to return contrast sensitivity, visual acuity, light aversion, ERG A- and B-wave peak amplitudes, pupil light reflex, optic nerve axon abundance, ipRGC soma size, and ipRGC melanopsin immunoreactivity to sham blast levels [2]. Similar results were obtained after raloxifene treatment post-repeated ocular blast injury [6]. It was observed that raloxifene helped redirect microglia away from their pro-inflammatory M1 activation state and toward a more protective M2 state in the context of TBI or repeated ocular blast injury [2,6]. What makes raloxifene an especially promising therapeutic agent is that it has already been FDA approved for osteoporosis treatment, leading to the possibility of simply also using it for the treatment of TBI-induced visual deficits [2]. Results suggest that greater doses of raloxifene may have a therapeutic effect on the optic nerve, as it reduces inflammation by improving the phagocytosis of axonal debris, thereby resulting in fewer injured axons [23].

Another pharmacological intervention being explored is the treatment of TBI with P7C3-S243, a neuroprotective agent [19]. A blast-induced TBI study in a mice found intraperitoneal injections of P7C3-S243, starting 5 min post-blast and continuing twice daily, to be effective in maintaining pre-blast PERG responses, suggesting maintained visual function, instead of the reduced PERG amplitudes seen in mice treated with only a DMSO vehicle [19].

Finally, a study showed that taurine, which is found in free form in neurons and serves as an important factor in brain growth and development, may also play a role in neuroprotection after TBI by preserving the neuronal ultrastructure, enhancing mitochondrial function, and modulating apoptosis-related protein expression [49].

## 7. Discussion

There are many post-mortem studies available that can help us determine the time scale of TBI. We now know that tissue degeneration mostly starts within 48 hours post-injury and can persist for up to a week or even longer [12].

The early signs of TBI cannot be assessed with ultrasonographic ONSD, as a certain post-injury period is needed for nerve thinning to occur. However, the literature does not provide us with a well-outlined post-injury time scale; therefore, we are unable to tell whether a method exists for both the early and late detection of TBI. Of course, early assessment is the most important. Further studies are needed to determine its usability, but it gives us a potential method for a detailed assessment of TBI.

Altogether, our focus was not to present a detailed clinical description of the disease but rather to highlight the possible routes for the visualization and implementation of new findings in the assessment. It is, however, important to reflect on some deficits in the information available in the selected articles. They are not able to answer some questions, including as to why the loss of the visual field was significantly more extensive in moderate-to-severe TBI (prevalence rate: 39.8%) than in mild TBI (6.6%). In contrast, deficits more closely related to central vision, including accommodative dysfunction (42.8%) and convergence insufficiency (36.3%), have less impact on the retina [50].

Also, the selected articles were not able to reflect the full scale of TBI variability, not to mention the available models. The differences in blast-related TBI [2,7,10,17,19,20,26,33,38,44], car accident-caused sTBI [51,52], or sports-related rmTBI [16,21,25,28,35,39,48] can be enormous. Here, we focused on the retinal effects and how these effects can lead us to the early diagnosis of TBI severity and not on the causes.

The importance of utilizing non-invasive ocular imaging of retinal blood flow, oxygen levels, or cells to assess TBI severity cannot be emphasized enough for early detection, as cerebral blood flow and metabolism are closely associated with retinal blood flow [9]. Live microglial cells can be visualized by the human eye label-free by employing adaptive optics [53]. This method bears the greatest potential in the identification of TBI severity without excessive harm and costs since studies suggest that microglia can be used as the first line of markers for TBI due to their obvious role in protection, including the induction of inflammation and the elimination of cellular debris [36]. The other glial elements in the retina, astrocytes, are the key players in retinal homeostasis, which is strongly affected by TBI. Avoiding the elimination of astrocytes possibly helps prevent the accumulation of pathological symptoms in the CNS [36], while astrocytes can also be imaged in the eye, together with ganglion cells in the GCL [54].

TBI interventions based on the application of surgical decompression or corticosteroids have shown limited success [12].

Treating the effects of TBI with antioxidant therapies might be limited due to the over-suppression of beneficial ROS. However, controlled ROS inhibition can help, like microglial inhibition with Minocycline, which lowers ROS level by suppressing microglial activation [12]. Also, ROS inhibition may have an efficacy time window that lasts only 3–6 h after the head trauma [12]. However, as indicated, arginase can compete with nitric oxide synthase to also limit oxidative stress [55].

Typical retinal changes detectable by OCT, such as the decrease in thickness of the RNFL or the RGC layer/inner plexiform layer, are generally not significant or measurable immediately after the acute injury. Instead, they tend to manifest weeks, months, or even years following the TBI event; therefore, OCT is not used to directly diagnose the acute TBI event. Instead, it is used to monitor and help determine the neurodegenerative consequences affecting the retina and optic nerve following TBI, particularly by identifying characteristic, chronically developing layer thinning patterns. Diagnosis in this context refers to identifying TBI-induced retinal/optic nerve damage based on these typical, chronically established structural changes [56].

Given the retina’s accessibility and shared pathology with the CNS, ophthalmic imaging techniques, such as OCT and pupillometry, have emerged as potential easily accessible and non-invasive diagnostic tools for assessing TBI severity. Studies indicate that alterations in RNFL thickness, RGC layer integrity, vascular abnormalities, and microglial activation due to inflammation may serve as quantifiable biomarkers for TBI-related neurodegeneration but not for primary diagnosis. These advancements underscore the potential for integrating retinal assessments into clinical protocols for TBI evaluation and management, but only some techniques show promise for the early diagnosis of TBI.

## 8. Conclusions

TBI exerts significant effects on the retina, characterized by neuroinflammatory responses, oxidative stress, structural degeneration, and functional impairments. The similarity between retinal and CNS pathophysiology highlights the applicability of retinal biomarkers in diagnosing and tracking TBI progression. Further interdisciplinary research is necessary to elucidate the mechanisms underlying TBI-induced retinal pathology and to develop targeted therapeutic strategies aimed at mitigating neurodegenerative sequelae.

To conclude, the impact of TBI on the visual system is multifaceted, ranging from structural degeneration to functional deficits. The extent of TBI depends on the type and severity of the injury. Both early and delayed imaging methods, such as DTI and OCT, can help in diagnosing and monitoring TBI-related visual impairments and TBI itself. More recently, because of the strong relationship between TBI and the caused visual impairments, we hypothesize that direct imaging of the eye with adaptive optics can be used to assess the level of TBI, based on the latest literature.

## Figures and Tables

**Figure 1 ijms-26-05171-f001:**
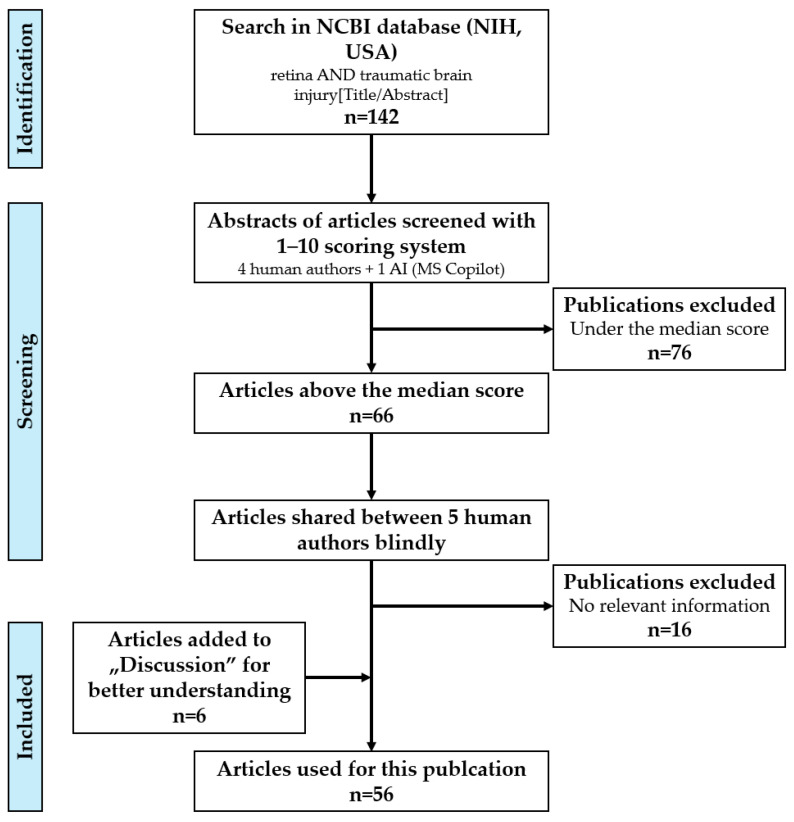
Flow diagram showing searches, screening, and selection for inclusion.

**Figure 2 ijms-26-05171-f002:**
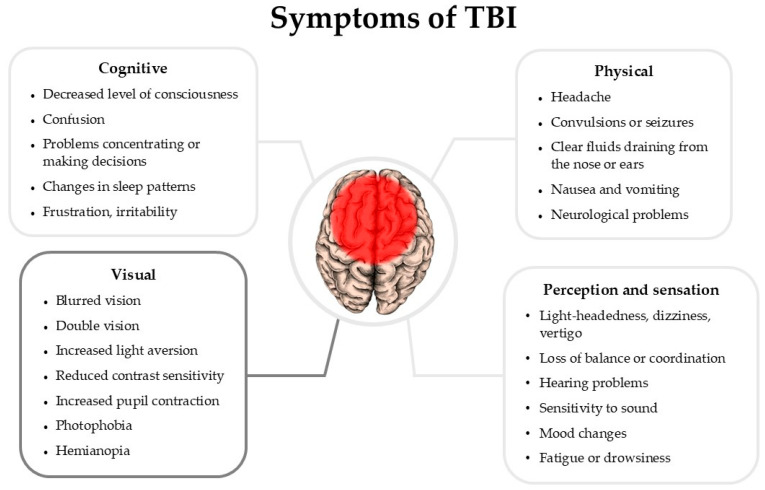
Symptoms of TBI in patients are described based on related references in this review article. The list of symptoms is based on NIH health information data [1,2,3,4,5].

**Figure 3 ijms-26-05171-f003:**
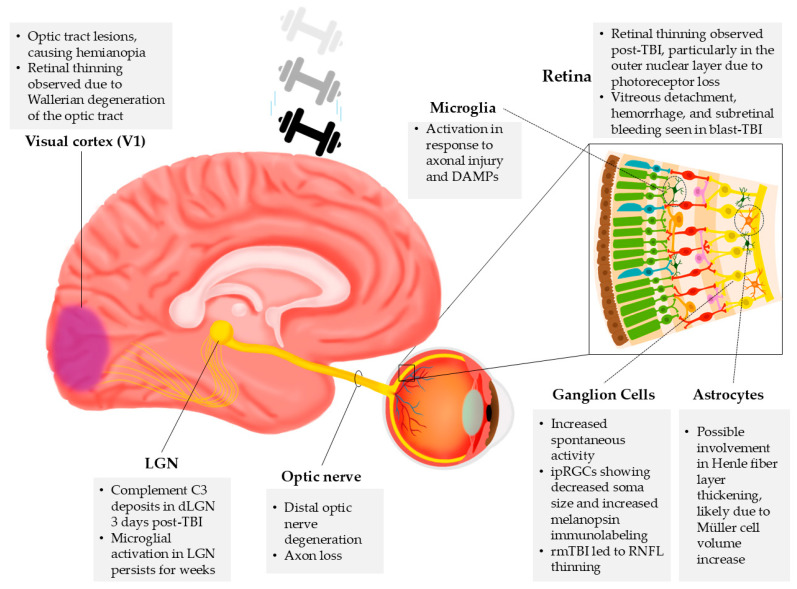
Effects of TBI on the visual system. According to the analyzed literature, different levels and centers in the visual system are affected by TBI, including the visual cortex (V1), LGN, optic nerve, and retina, as well as the following cell types: microglia, ganglion cells, and astrocytes.

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
