# Peer review of "Eyes Are the Windows to the Soul: Reviewing the Possible Use of the Retina to Indicate Traumatic Brain Injury"

_ijms, 2025, doi:10.3390/ijms26115171_

Round 1

Reviewer 1 Report

Comments and Suggestions for Authors

This review article entitled “ reviewing the possible of retina to indicate TBI. Authors have made a comprehensive review of recent literature. However, there are several concerns of this manuscript before considering being published.

  1. Since the retina is the main point to be discussed with TBI, the oculomotor function deficit, such as diplopia, is not the point. Authors should provide the clinical incidence of sensory visual dysfunction after TBI in the introduction first.
  2. There are many different kinds of TBI models. Authors should provide what kinds of animal model or lesion area in clinical TBI will induce the retinal and optic nerve lesions.
  3. L78 and Fig.1 elevated pupil constriction after TBI is a confusing expression. The reference 16 did not describe this pupil light reflex, please revise
  4. L151, only the distal part of the ON was damaged, authors should provide the reason why the proximal part of ON and RGCs will be intact. Is this caused by the specific animal model of TBI?
  5. Line150-157 In this paragraph, authors accumulated several different TBI models. It is inappropriate to put it together as a an uniform presentations of TBI.
  6. Figure 2. Visual cortex lesion may not induce retinal thinning and optic tract lesion, unless a trans-synaptic degeneration occurs afterTBI. Usually, the ON axon lesion not only induce the distal part axonopathy, but also the proximal part of axon and RGCs death.
  7. Line 297-299 provocative PERG is different from standard PERG. This is a confusing expression.
  8. Line375-377 and Line 394-396 These two paragraphs are repetitive
  9. Line 422-424 Need to add the reference

Author Response

Dear Editors, Dear Reviewers

Thank you for taking the time to review our paper and thank you for your suggestions.

Below, we included a point-by-point response to all of the notes made by the Editors and the Reviewers (in italics). 

This review article entitled “ reviewing the possibility of retina to indicate TBI. Authors have made a comprehensive review of recent literature. However, there are several concerns of this manuscript before considering being published.

  1. Since the retina is the main point to be discussed with TBI, the oculomotor function deficit, such as diplopia, is not the point. Authors should provide the clinical incidence of sensory visual dysfunction after TBI in the introduction first.

Thank you for the comment. We agree that we needed to add this relevant information. As described in the Methods section, we looked into filtered information only in the main body of the text. ‘...142 articles in the first round of search. Following this, we used a 1-10 scoring system (1-3: non-relevant, 4-6: somewhat relevant, 7-9 very relevant, 10 perfect match to the topic) on the titles and abstracts. Four authors independently scored all articles based on the titles and abstracts. We used the Copilot (Microsoft) as a fifth scorer’. After this primary selection, the human authors read a highly scored selection only from these. Based on this, we were not able to get every detail, including the incidence rates. We were focusing on the available information from this selection. In the discussion, however, we added extra papers to further improve the embeddedness of our review into the literature. We do not want to interfere with this concept, so we added the related piece of information to the discussion only. We added a systematic review and meta-analysis, offering visual prevalence data (Merezhinskaya, N., et al. (2019). Visual Deficits and Dysfunctions Associated with Traumatic Brain Injury: A Systematic Review and Meta-analysis. Optometry and Vision Science, 96(8), 535-548.) to highlight the links more in the discussion.

  1. There are many different kinds of TBI models. Authors should provide what kinds of animal model or lesion area in clinical TBI will induce the retinal and optic nerve lesions.

As mentioned before, there are some meta-analyses available to reflect the clinical prevalence. (Merezhinskaya, N., et al. (2019). Visual Deficits and Dysfunctions Associated with Traumatic Brain Injury: A Systematic Review and Meta-analysis. Optometry and Vision Science, 96(8), 535-548.). The available animal models in the selected articles are not fully described by our selection of articles (described in the selection algorithm). The selected articles were not able to reflect the full scale of TBI variability, not to mention the available models. The differences in blast-related TBI (add REF.), car accident-caused sTBI (REF), or sports-related rmTBI (REF) can be enormous. Here, we focused on the retinal effects and how these effects can lead us to the early diagnosis of TBI severity and not on the causes. (Also added in the discussion).

  1. L78 and Fig.1 elevated pupil constriction after TBI is a confusing expression. The reference 16 did not describe this pupil light reflex, please revise. 

Thank you for the comment. We changed it to “enhanced” pupil constriction in the text and added the right references to the figure. 

  1. L151, only the distal part of the ON was damaged, authors should provide the reason why the proximal part of ON and RGCs will be intact. Is this caused by the specific animal model of TBI? 

In this study, only the distal optic nerve was damaged because the TBI model used by Wang et al. (2013), central fluid percussion injury model, induces axonal injury near the optic chiasm instead of more proximally near the ganglion cell layer.  This model spares the proximal segment and RGCs by avoiding immediate mechanical disruption, unlike crush or transection models.  We clarified this in the text (Section 4, Paragraph 3).

  1. Line 150-157: In this paragraph, authors accumulated several different TBI models. It is inappropriate to put it together as a uniform presentation of TBI.

We thank the reviewer for this comment. We added a sentence for clarification in the first paragraph (L148): ‘The literature describes different TBI types and models. For clarification, we describe the models or type of TBI, if needed, as they can have minor but important differences in their effects on the visual system.’

  1. Figure 2. Visual cortex lesion may not induce retinal thinning and optic tract lesion, unless a trans-synaptic degeneration occurs after TBI. Usually, the ON axon lesion not only induces the distal part axonopathy, but also the proximal part of axon and RGCs death.

In Figure 2 (‘Effects of TBI in the visual system. Based on the processed literature, different levels and centers are affected by traumatic brain injury in the visual system, including Visual cortex (V1), Lateral geniculate nucleus (LGN), Optic nerve, Retina -including cell types: Microglia, Ganglion cells, and Astrocytes.’), we also described the secondary effects due to primary insult. As previously described by multiple papers, the primary lesion can induce retinal thinning through secondary (Wallerian) degeneration. We focused on the retinal effects. Here, when a lesion occurs in the visual cortex (e.g., due to stroke or TBI), the neurons in this area are damaged or die. These cortical neurons are postsynaptic to neurons originating in the LGN, which in turn are postsynaptic to RGCs. The loss of trophic support and normal synaptic activity from the damaged cortical neurons can trigger a degenerative cascade that travels retrogradely across the synapses, eventually leading to the atrophy and death of RGCs and the thinning of their axons. (DOI: 10.1098/rspb.2018.2733)

  1. Line 297-299 provocative PERG is different from standard PERG. This is a confusing expression.

They are indeed different approaches. Thank you for drawing our attention to it. We have further clarified the end of the sentence as follows: ‘Pattern electroretinography (PERG) may be a useful tool as it can assess RGC function non-invasively. In fact, it has previously been shown that marked PERG signal changes occur in mice following blast-induced TBI.  It appears that performing the PERG measurements under conditions of increased ocular pressure resulted in differences in retinal function that can be detected with PERG measurements.’

  1. Line 375-377 and Line 394-396: These two paragraphs are repetitive 

Thank you for the suggestion. We have moved this paragraph to the end of the section, but somehow we did not manage to delete it. Now only the second remained in the text (L398).

  1. Line 422-424 Need to add the reference (https://doi.org/10.3389/fnins.2023.1021152) [11]

Thank you for your notice. We added the reference (L449) and with the changes in the text it is now reference [12].

We added a PDF files with formatted answers for both reviewers, please find it attached to our revised manuscript.

We thank both reviewers for their extensive work and help in correcting our article. We hope that with the changes, it will now satisfy the requirements for publishing.

Reviewer 2 Report

Comments and Suggestions for Authors

This is an extensive review on retinal markers and findings in TBI, with interesting summary of the literature 

Methods

It is interesting to see how the authors incorporated AI in the review. Did they not read the articles that were assigned to the AI model? It is not clear how they shared the articles. What does X articles were added to the discussion mean? Please explain 

Section about manifestations 

can you organize with the table first and the symptoms. That would make it easier to read 

Effects of TBI on the visual system

Clinicians usually try to sort out effects of TBI,  with the easiest to get test first . In my opinion that is visual acuity  and  imaging tests (OCT , RNFL , GCL analysis, MRI, PET...).than Visual fields and other psychophysical and electrodiagnostic tests.  Why start this section with discussion of  pattern ERG? This is a very difficult test to obtain in normals, One needs a well trained technician who can obtain reliable and repeatable  results.  Therefore pattern ERG are typically done in only a few labs and most of those are in Europe. I can only imagine to do a pattern ERG study in patients with TBI and severe photophobia. VEPs and flash ERG PhNR responses may be easier to obtain. 

Line 163  What has the conditions of elevated ocular pressure to do with this? You are studying TBI and not traumatic glaucoma. Maybe a human author  can clarify and organize this section to make it easier to read with some subdivisions . 

The readers will be most interested in a paragraph on RNFL /GCL OCT ; this is a test available in every eyecare providers office and easy to obtain on TBI patients  

Same is true for the next section . Needs some editing; line 253 Push the CNS " beyond the limit " is spoken English, please rephrase.   Table 1   lay out is poor and the columns are not separated

Translational Implementation , you discuss Non accidental trauma,blast injury, mice experiments and anti-LPA antibodies , anakinra,  Raloxifen and Amyloidosis all in one section . Please organize and make subheadings. 

make more sections or have a table . line 334 what is a vehicle treatment group ?

 . 

Author Response

Dear Editors, Dear Reviewers

Thank you for taking the time to review our paper and thank you for your suggestions.

Below, we included a point-by-point response to all of the notes made by the Editors and the Reviewers (in italics).

This is an extensive review on retinal markers and findings in TBI, with interesting summary of the literature. 

Methods

It is interesting to see how the authors incorporated AI in the review. Did they not read the articles that were assigned to the AI model? (1) It is not clear how they shared the articles. What does X articles were added to the discussion mean? (2)  Please explain. 

  • Thank you for your question. We used AI only for the scoring part of the process, then divided the resulting articles among human authors for reading. We made a flow diagram (Fig.1)  to make the process more understandable.
  • “X article were added”  means that in addition to the 142 articles found initially, we have added 2 other articles for better understanding. (Ref. 50; Ref. 51) Thank you for the comment; it really was not clear, but we now clarified it in the text (L66).

Section about manifestations 

Can you organize with the table first and the symptoms. (3)  That would make it easier to read. 

  • We put Figure 2. at the beginning of the section to help readability.

Effects of TBI on the visual system

Clinicians usually try to sort out effects of TBI,  with the easiest to get tested first . In my opinion that is visual acuity  and  imaging tests (OCT , RNFL , GCL analysis, MRI, PET...) rather than visual fields and other psychophysical and electrodiagnostic tests.  Why start this section with discussion of  pattern ERG? (4) This is a very difficult test to obtain in normals, one needs a well trained technician who can obtain reliable and repeatable  results.  Therefore pattern ERG are typically done in only a few labs and most of those are in Europe. I can only imagine to do a pattern ERG study in patients with TBI and severe photophobia. VEPs and flash ERG PhNR responses may be easier to obtain. 

  • We agree with the reviewer. The easier the better. None of the writers are clinicians. All of us are neuroscientists, therefore, it is important to assess the manuscript by clinician experts, which we lacked up to this point (as both of the reviewers seem to have a clinical focus that helped a lot). TBI patients will be tested for visual effects a bit later; some of them are not even in a state where they can be tested (severe trauma, open head trauma). All of the mentioned tests (OCT , RNFL , GCL analysis, MRI, PET) and the related studies lack the detail (no cellular data is available, only the thickness of layers and a rough estimation of microstructural changes) and therefore the possibility of early diagnosis is not available, as these changes develop in days or even weeks.

For example, in TBI diagnosis, OCT primarily serves to detect damage to the retina and optic nerve much later following TBI, rather than to acutely diagnose the brain injury itself. Retinal and optic nerve changes measurable by OCT can only serve as markers of secondary neurodegenerative processes occurring after TBI.

Typical retinal changes detectable by OCT, such as the decrease in thickness of the retinal nerve fiber layer (RNFL) or the ganglion cell layer/inner plexiform layer (GCL/IPL), generally do not become significant and measurable immediately after the acute injury. Instead, they tend to manifest weeks, months, or even years following the TBI event, therefore, OCT does not directly diagnose the acute TBI event. Instead, it is used to monitor and help diagnose the neurodegenerative consequences affecting the retina and optic nerve following TBI, particularly by identifying characteristic, chronically developing layer thinning patterns. The "diagnosis" in this context refers to identifying TBI-induced retinal/optic nerve damage based on these typical, chronically established structural changes (10.1089/neu.2021.0182) (We added this section to the Discussion).

We agree with the reviewer (also mentioned by the other reviewer). PERG is now in the middle of the section, and we also highlighted that it only shows a reliable outcome in normalized conditions (e.g., angle of the head is the same). In a real-life setting VEP and flash ERG have possible usability but with the same limitations as mentioned before.

Line 163: What has the conditions of elevated ocular pressure to do with this? (5) You are studying TBI and not traumatic glaucoma. Maybe a human author can clarify and organize this section to make it easier to read with some subdivisions . 

(5) Dutca et al. (2014) performed PERG measurements while the TBI mice were tilted in a 60-degree head down position to cause increased intraocular pressure and found these results to differ from those obtained from neutral position PERG assessment of TBI mice. We clarified this in the text (Section 4, Paragraph 4), but we do not think subheadings are needed, to avoid overly complicated divisions in the text.

The readers will be most interested in a paragraph on RNFL /GCL OCT ; this is a test available in every eyecare providers office and easy to obtain on TBI patients.  Same is true for the next section . Needs some editing; line 253 Push the CNS " beyond the limit " is spoken English, please rephrase. (6)  Table 1   lay out is poor and the columns are not separated (7)

(6) Changed to: The activation of inflammatory pathways and the disruption of barriers (BBB and BRB) overwhelms the CNS and induces secondary processes(Section 5, Paragraph 7). We added a paragraph about OCT at the end of the discussion.

(7) Thank you for your comment. We made the columns more separated.

Translational Implementation 

ou discuss Non accidental trauma, blast injury, mice experiments and anti-LPA antibodies, anakinra, Raloxifen and Amyloidosis all in one section. Please organize and make subheadings. (8) ake more sections or have a table (9). Line 334 what is a vehicle treatment group? (10)

(8/9) We reorganized this section to have the following subheadings: i) The importance of early TBI detection, ii) Methods for early TBI detection, and iii) Possible interventions for early TBI treatment. (Section 6)

(10) The vehicle treatment group was a control group that received DMSO vehicle injections after blast-induced TBI instead of P7C3-S243 injections.  We clarified this in the text. (Section 6, Subheading 3, Paragraph 5).

We added a PDF file with formatted answers for both reviewers. Please find it attached to our revised manuscript.

We thank both reviewers for their extensive work and help in correcting our article. We hope that with the changes, it will now satisfy the requirements for publishing.

Round 2

Reviewer 1 Report

Comments and Suggestions for Authors

Thanks to the authors for making extensive revisions; no further comments

Author Response

Rev1: Thanks to the authors for making extensive revisions; no further comments

Reply: Thank you for taking the time to review our article.

Reviewer 2 Report

Comments and Suggestions for Authors

The authors have improved the manuscript satisfactorily. They have clarified their use of AI and have provided an extra " Figure 1": this  flow chart is a nice addition to explain the methods.

 They did move Figure 2 to the beginning of the discussion of manifestations : this makes it easier to follow . 

Figure 3 also deserves to be seen before one reads the entire Effects of TBI on the Visual System . If possible please move up Figure 3.  

Section 4; Thank you for clarifying the mouse PERG and the impact of positioning on IOP.

Table 1 middle column , spacing is off and not aligned with first column.  Please align.

Possible translational implementation of retinal Markers in TBI

Methods for early TBI detection:  Th authors still list the PERG first and have the statement about increased ocular pressure. I would have preferred discussion of visual fields and imaging first, conform with clinic flow in neuro-ophthalmology , I now realize that the authors are neuro-scientists and the readers will be neuroscientists not clinicians.   

Author Response

(Reviewer comments are in bold, answers are in italics!)

The authors have improved the manuscript satisfactorily. They have clarified their use of AI and have provided an extra " Figure 1": this  flow chart is a nice addition to explain the methods.

 They did move Figure 2 to the beginning of the discussion of manifestations : this makes it easier to follow. 

Thank you for the revision of our changes. We are now providing a re-revised version of our manuscript focusing on the changes suggested by the reviewer here. Our answers are in italics.

Figure 3 also deserves to be seen before one reads the entire Effects of TBI on the Visual System . If possible please move up Figure 3.

Thank you for the suggestion. We moved Figure 3 to the beginning of the section. 

Section 4; Thank you for clarifying the mouse PERG and the impact of positioning on IOP.

Thank you for the suggestion again.

Table 1 middle column , spacing is off and not aligned with first column.  Please align.

We have aligned the columns to correct the spacing. (Also, please note that the final processing of tables will be done by the journal for the webpage and the PDF version.)

Possible translational implementation of retinal Markers in TBI

Methods for early TBI detection:  The authors still list the PERG first and have the statement about increased ocular pressure. I would have preferred discussion of visual fields and imaging first, to conform with clinic flow in neuro-ophthalmology , I now realize that the authors are neuro-scientists and the readers will be neuroscientists not clinicians.

We moved the first instance of PERG later in the last round; we now moved this part later in the section and added a clarification as requested. We wanted to highlight the fact that PERG is hard to standardise and even ocular pressure changes can have altered outcomes. -We added “However, under conditions of elevated ocular pressure, achieved by tilting the TBI mice to a 60-degree head down position during PERG testing, suggesting alteration of results due to measurement changes,…-that is basically consistent with the reviewer’s opinion.

We hope that with the changes we made, our article now meets the requirements. We thank the Reviewers for their hard work and the Editors for their help.
